# One-Stage Hydrothermal Growth and Characterization of Epitaxial LaMnO_3_ Films on SrTiO_3_ Substrate

**DOI:** 10.3390/ma15175928

**Published:** 2022-08-27

**Authors:** Keyu Guo, Yaqiu Tao, Yunfei Liu, Yinong Lyu, Zhigang Pan

**Affiliations:** College of Materials Science and Engineering, Nanjing Tech University, Nanjing 211800, China

**Keywords:** LaMnO_3_, hydrothermal, epitaxy, thin film, interface, strain

## Abstract

Epitaxial LaMnO_3_ thin films were grown on SrTiO_3_ substrate using a one-stage hydrothermal route from La(NO_3_)_3_, MnCl_2_ and KMnO_4_ in an aqueous solution of 10 M KOH at 340 °C. Scanning electron microscopy (SEM) and energy dispersive X-ray spectroscopy (EDS) indicate full coverage of LaMnO_3_ on the substrate. X-ray diffraction in the symmetric ω/2θ mode suggests the film has an out-of-plane preferred orientation along the [001] direction of the substrate. The LaMnO_3_ epitaxial thin film growth mechanism is proposed based on the analysis of the atomic sharp interface formed between LaMnO_3_ and the SrTiO_3_ substrate, as seen by aberration−corrected scanning transmission electron microscopy (AC−STEM) imaging in combination with electronic energy loss spectroscopy (EELS). Compared with the conventional vapor deposition methods, the one-stage hydrothermal route opens up a new way to fabricate complex oxide epitaxial heterostructures.

## 1. Introduction

Lanthanum manganite perovskite (LaMnO_3_) and derivatives are appealing complex oxides for their functional applications in high-temperature solid oxide fuel cells (SOFCs) [1,2,3], supercapacitors [4,5,6], volatile organic compound (VOC) elimination [7,8], photocatalysis [9] and spintronic devices [10]. In particular, LaMnO_3_ films have been subjected to intensive research as a highly competitive material for information storage since the discovery of the colossal magnetoresistance effect [11,12,13,14]. Bulk LaMnO_3_ displays an orthorhombic (Pnma; *a*= 5.6792 Å, *b* = 7.7030 Å, *c* = 5.5410 Å) or a trigonal ($R\bar{3}c$; *a*= 5.6792 Å, *b* = 7.7030 Å, *c* = 5.5410 Å) symmetry at room temperature depending on the substitution or synthesis conditions [15,16]. Both Pnma and $R\bar{3}c$ are subgroups of the space group of idealized LaMnO_3_ ($Pm\bar{3}m$; *a* ≈ 3.90 Å) [17]. The idealized bulk LaMnO_3_ itself possesses only high-spin Mn^3+^ ($t_{2g}^{3}e_{g}^{1}$), resulting in an antiferromagnetic insulator. When Mn^4+^ cations are introduced to LaMnO_3_ via either divalent cation substitution for La^3+^ cations or annealing in an oxygen atmosphere, lanthanum manganite exhibits ferromagnetism, which can be explained by the double exchange mechanism [18]. In addition to doping or heat treatment, growing LaMnO_3_ thin films on substrates has been an efficient way to manipulate the physical properties of LaMnO_3_. Symmetry breaking at the interface is mainly the source of novel properties. Interfacial mismatches in lattice parameters, oxygen octahedral torsions and tilts, chemical composition or valence states induce emerging physical and chemical properties that are distinct to the bulk LaMnO_3_.

When antiferromagnetic LaMnO_3_ forms a heterostructure with paramagnetic LaNiO_3_, electrons are transferred from LaMnO_3_ to LaNiO_3_, leading to magnetism in LaNiO_3_ [19]. For atomic layer superlattices of LaMnO_3_(*n*)/SrTiO_3_(*n*), with *n* being the number of corresponding layers, the magnetization in LaMnO_3_ was enhanced when *n* = 8, while *n* = 1 or 2 superlattices showed decreased magnetization. LaMnO_3_ thin films tend to be ferromagnetic insulators when grown on SrTiO_3_ substrate [20]. The La^3+^ and O^2-^ stoichiometries can be improved by using a reducing atmosphere during the growth process, and therefore, the bulk-like electronic and magnetic state of LaMnO_3_ can be stabilized [21].

While most heterostructures involving LaMnO_3_ are fabricated using vapor deposition methods, such as pulsed laser deposition (PLD) [22,23,24], physical vapor deposition (PVD) [25], chemical vapor deposition (CVD) [26] and molecular beam deposition (MBE) [27,28], we have been working on developing hydrothermal routes to synthesize LaMnO_3_ thin films on appropriate substrates. Hydrothermal routes have intrinsic advantages of low temperature, low cost and the ability to control the grain size and polymorph of the target product compared with the well-developed vapor methods [29]. Our one-stage hydrothermal approach to perovskite oxides differs from the conventional chemical solution deposition (CSD) method, which involves the deposition of precursors on the substrate, followed by heat treatment to induce pyrolysis and subsequent nucleation and growth of desired perovskite [30]. Nano islands are usually formed on substrates in CSD. On the contrary, in the one-stage hydrothermal route, high-quality thin films can be obtained directly by choosing appropriate hydrothermal conditions. T. Morita’s group successfully grew PbTiO_3_ epitaxial films with piezoelectric properties on the (001) surface of single-crystal SrTiO_3_ substrates [31]. High-quality BiFeO_3_ multiferroic single-crystal films can also be deposited onto SrTiO_3_ single-crystal substrates using the hydrothermal method [32,33]. Despite the success of applying hydrothermal routes to produce a variety of perovskite thin films on substrates, growing LaMnO_3_ thin films using a one-stage hydrothermal route is rarely reported. 

Single-crystal SrTiO_3_ ($Pm\bar{3}m$; *a* = 3.905 Å) provides a suitable lattice match to LaMnO_3_ and was chosen as the substrate for LaMnO_3_ film growth. In this work, we prepared LaMnO_3_ thin films on a [001] SrTiO_3_ substrate under hydrothermal conditions. Epitaxial LaMnO_3_ film was obtained, and the film growth mechanism was investigated. We evaluated the quality of the LaMnO_3_ thin film and the interfacial structure using a combination of X-ray diffraction, surface morphology and scanning transmission electron microscopy.

## 2. Materials and Methods

### 2.1. Synthesis of Bulk LaMnO_3_ Powders 

Prior to LaMnO_3_ film growth, hydrothermal conditions for preparing bulk LaMnO_3_ powders were explored in terms of precursor type, temperature, mineralizer concentration and length of time for hydrothermal treatment. In this work, La(NO_3_)_3_·6H_2_O (99.0%, Aladdin Biochemical Technology, Shanghai, China) was used as a lanthanum precursor, and MnCl_2_·4H_2_O (99.0%,Shanghai Macklin Biochemical, Shanghai, China) and KMnO_4_ (99.5%, Shanghai Lingfeng, Shanghai, China) were used as manganese precursors. KOH (85.0%, Sinopharm, Beijing, China) served as a mineralizer. Solutions of 0.05 M KMnO_4_, 0.2 M MnCl_2_·4H_2_O and 0.25 M La(NO_3_)_3_·6H_2_O were prepared with deionized water before the synthesis. First, 20 mL of KMnO_4_ solution and 20 mL of La(NO_3_)_3_ solution were mixed, and then 44.88 g of KOH was added in several portions to the above mixture. After cooling to room temperature, the resultant mixture was mixed dropwise with 20 mL of MnCl_2_ solution. The resulting brown suspension was diluted by adding deionized water to a total volume of 70 mL and transferred to the Hastelloy autoclave (100 mL in volume) for the subsequent hydrothermal treatment. Powder samples were filtered after a certain time period during the hydrothermal reaction at 340 °C. The final product was washed with deionized water and dried with ethanol.

### 2.2. Preparation of LaMnO_3_ thin Films on SrTiO_3_ Substrate

LaMnO_3_ thin films were prepared using a similar procedure, except a SrTiO_3_ substrate (5.0 × 5.0 × 0.5 mm^3^) was placed at the bottom of the autoclave before the hydrothermal treatment. The film obtained after a period of 2 h was subject to ultrasonic cleaning for 10 min, and washed with deionized water and ethanol. 

### 2.3. Material Characterization

The morphology of the thin film was characterized by a field emission scanning electron microscope (FESEM; Ultra-55, Zeiss, Oberkochen, Germany). X-ray diffraction patterns were recorded using a Rigaku Smartlab diffractometer (Cu target, 40 kV and 100 mA). Powder diffraction data for bulk LaMnO_3_ powders were collected in the conventional Bragg Brentano mode using a linear position-sensitive detector covering 6° in 2θ for phase quantitative analysis. For LaMnO_3_ thin films, X-ray diffraction data were obtained in the symmetric ω/2θ mode using a 2-bounce Ge(220) monochromator and a scintillation detector. The STEM sample was prepared by a focused ion beam (FIB; Crossbeam 350, Zeiss, Germany) using gallium ions, and mounted onto the Ominiprobe half grid for analysis. STEM images were recorded by a double Cs-corrected JEOL ARM-300CF electron microscope equipped with a cold field emission gun. ADF images were recorded with the inner and outer collector semi-angle settings at 38 and 153 mrad, respectively. The locations of the A-site and B-site atomic columns were obtained by Gaussian fitting in the CalAtom software package. The displacement vector was calculated by the deviation of the B-site atom from the ideal mass center of surrounding A-site atoms. Electron energy loss spectroscopy was used to analyze the elemental distribution at the interface, and the strain at the interface was calculated in Digital Micrograph software using the GPA plug-in.

## 3. Results and Discussion

Figure 1 shows the powder X-ray diffraction patterns of samples obtained after a period of 2, 3, 6, 12 and 18 h under hydrothermal treatment. The low background in the diffraction patterns indicates adequate crystallinity of the bulk powders. As shown in Figure 1, the samples at 2 h and 3 h contain impurities, namely, K-rich birnessite and La(OH)_3_.

For the product at 6 h, a small amount of La(OH)_3_ can be identified, and no reflections due to K-rich birnessite are observed. No reflections from La(OH)_3_ or K-rich birnessite are detected in the sample after a period of 12 h. The powder X-ray diffraction patterns for the samples at 12 and 18 h indicate the presence of the trigonal phase of LaMnO_3_. Since most reflections due to the orthorhombic phase coincide with the reflections from the trigonal phase, the identification of the orthorhombic phase cannot be made unambiguously. Rietveld quantitative analyses including both trigonal and orthorhombic phases of LaMnO_3_ were then carried out using the GSASII software package [34] in order to evaluate the phase composition at each stage of the hydrothermal process. In Rietveld refinement, the instrumental parameters were obtained using the standard reference material LaB_6_ (SRM-660b) from the National Institute of Standards and Technology (NIST) and remained the same during Rietveld refinement. Lattice parameters and scale factors were involved in the refinement while atomic positions or thermal displacement parameters remained fixed. Anisotropic peak broadening was employed to interpret the sample micro strain effects. The powder sample at 2 h was not included in Rietveld refinement because no satisfying refinement can be obtained due to the lack of a proper model to describe the strong preferred orientation of the sample. The final Rietveld phase quantitative refinement of the sample at 18 h is shown in Figure 2 as a representative, and other Rietveld plots are in Appendix A. The phase compositions for samples at 3, 6, 12 and 18 h are listed in Table 1.

As indicated by the trigonal to orthorhombic phase ratio listed in Table 1, the trigonal phase dominates in the early stage of the hydrothermal synthesis. Previous research suggested that the trigonal phase can be stabilized by improving the Mn^4+^ content in LaMnO_3_ [35]. In our case, KMnO_4_ is used as a manganate precursor and Mn^4+^ is ready to be incorporated into LaMnO_3_ at the early stage of synthesis. As the synthesis proceeds, the amount of available Mn^4+^ decreases, resulting in a relatively more orthorhombic phase being formed.

Powder samples were further examined by SEM, and images for samples at 2, 3, 6 and 12 h are shown in Figure 3. Cube-shaped LaMnO_3_ was formed together with rod-shaped La(OH)_3_ at 2, 3 and 6 h, and the amount of La(OH)_3_ decreased gradually in the process of hydrothermal synthesis. The K-rich birnessite phase was not directly observed, but EDS shows that potassium was uniformly distributed across the sample surface (Appendix A). Pure LaMnO_3_ was yielded at 12 h. 

After successfully obtaining LaMnO_3_ powders using the one-stage hydrothermal method, similar hydrothermal conditions were employed to prepare LaMnO_3_ films on SrTiO_3_ substrate. The lanthanum and manganese precursor concentrations were half of those used for the synthesis of bulk LaMnO_3_ powders and the film growth was carried out for a period of 2 h. The obtained LaMnO_3_ films on the SrTiO_3_ substrate were characterized by SEM/EDS, shown in Figure 4a. SEM images suggest that the film covered the substrate completely and EDS shows that La, Mn and O elements are uniformly distributed across the surface. High-resolution X-ray diffraction was employed to characterize the LaMnO_3_ film using a two-bounce Ge(220) monochromator. A symmetric ω/2θ scan of the LaMnO_3_ film shows only (*00l*) peaks. For comparison, a symmetric ω/2θ scan of the substrate was also obtained under the same instrumental conditions. Despite the similar (*00l*) peak positions from the LaMnO_3_ film and the substrate, the difference in the full width at the half maximum (FWHM), as shown in Figure 4b,c, indicates that pseudo-cubic LaMnO_3_ films with an out-of-pane preferred orientation were obtained on the SrTiO_3_ substrate.

To further demonstrate the film orientation with respect to the substrate, the interfacial structure was examined using a double-spherical differential correction electron microscope. Figure 5a shows the atomic arrangement at the interface between the LaMnO_3_ film and the SrTiO_3_ substrate, confirming that the epitaxial LaMnO_3_ film was grown on the substrate. Figure 5b shows the cross-section of the LaMnO_3_ film along the [100]_pc_ (pc denotes pseudo-cubic) direction. In the ADF-STEM images, the contrast is approximately proportional to Z^1.67^, with Z being the atomic number. Therefore, the brightest atom columns correspond to the position of the heaviest element, i.e., lanthanum (Z = 57). Strontium (Z = 38) columns are formed by the second brightest features, which are in the horizontal line formed by La columns shown in Figure 5b. Mn and Ti columns are recognized as those in between La columns in LaMnO_3_ and Sr columns in SrTiO_3_. The O columns are largely invisible in the ADF mode. EELS mapping confirmed the assignment as discussed in the following section. Figure 5c displays a pixel intensity line scan of one row of atoms in the horizontal direction (white rectangle in Figure 5b) to differentiate the atoms at the interface. ADF-STEM imaging reveals that an atomic flat and sharp interface formed between LaMnO_3_ films and the SrTiO_3_ substrate.

Once the epitaxial heterostructure was identified, geometrical phase analysis (GPA) was performed to understand the lattice distortion of the LaMnO_3_ film. A wider field of view of the ADF−STEM images was chosen for GPA. The warm coloration on GPA represents a lattice expansion relative to the reference that was chosen from the core region of the substrate, while the cool coloration indicates that the lattice is contracted. If the film conforms to the substrate interplanar distance via epitaxy, there should be no obvious warm or cool coloration region on GPA. As shown in Figure 6a, the contrast between the substrate and the film in the in-plane direction of the film is relatively similar, suggesting no obvious lattice distortion along the [100]_pc_ direction. On the contrary, Figure 6b exhibits a significant large out-of-plane lattice spacing of the LaMnO_3_ film, as indicated by the warm coloration on the LaMnO_3_ side. The average lattice spacing d_001_ for SrTiO_3_ and LaMnO_3_ is 3.96 and 4.01Å, respectively. The average lattice spacing d_001_ is slightly larger than the ideal bulk LaMnO_3_ (≈3.90 Å). GPA suggests that the LaMnO_3_ film is in a tensile strain state.

Lattice strain affects the orbital ordering and magnetism at the interface between LaMnO_3_ and SrTiO_3_. In the meantime, B-site atomic displacement also has a profound influence on the physical properties of the LaMnO_3_ film. Therefore, the B-site atomic displacements for both LaMnO_3_ films and the SrTiO_3_ substrate were investigated. The arrows shown in Figure 7a represent the magnitude and direction of the displacement of the B-site atoms from the ideal positions, and the color indicates the coherent direction of the B-site displacement. 

The polar plot of the B-site atomic displacement (Figure 7b) shows that the direction is not restricted to any particular direction but is distributed over a wide range. The majority of the B-site displacement falls in the range of 2.5–10 pm and −30–120°. The B-site displacement analysis suggests the polarity of the LaMnO_3_ film. 

To further investigate the LaMnO_3_ film growth mechanism, the elemental distribution at the interface was analyzed using electronic energy loss spectroscopy. Figure 8 shows an atomic resolution of the ADF−STEM images, along with EELS mapping. As shown in Figure 8, Mn, Ti, La and Sr atoms have diffusion to varying degrees in between the LaMnO_3_ film and the SrTiO_3_ substrate, indicating a certain degree of cation intermixing at the interface. 

The cation diffusion may relate to the strong alkaline environment that causes some corrosion of the substrate to form Sr^2+^ vacancies. The resultant vacancies may be substituted by La^3+^ and therefore become active sites for nucleation and growth of LaMnO_3_ thin films. The free Sr^2+^ due to corrosion in the vicinity of the substrate may be incorporated into the newly formed LaMnO_3_ films. The diffusion of La and Sr is easily identified in the La and Sr maps shown in Figure 8. Furthermore, Figure 5b shows a bright feature of one atom column (middle) on the SrTiO_3_ side, which also indicates that La^3+^ is incorporated into the SrTiO_3_ substrate. Figure 9 shows a schematic view of the growth mechanism of LaMnO_3_ films. Mn^3+^ and Ti^4+^ have a weaker diffusion compared with La^3+^ and Sr^2+^, as seen in the Mn^3+^ and Ti^4+^ maps in Figure 8. Meanwhile, the O atomic columns on both sides of the interface are not as well-defined as the other elements, indicating the presence of octahedral distortions at the interface.

## 4. Conclusions

In summary, bulk LaMnO_3_ powders were successfully prepared by choosing appropriate lanthanum and manganese precursors under hydrothermal conditions. Trigonal LaMnO_3_ formation dominates the early stage of synthesis when a large amount of high valent manganese is available, while orthorhombic LaMnO_3_ has a preference over trigonal LaMnO_3_ at later stages. Epitaxial LaMnO_3_ film on SrTiO_3_ can be obtained using the hydrothermal route with growth parameters obtained from the bulk LaMnO_3_ powder synthesis. A highly out-of-plane preferred orientation of the LaMnO_3_ film was revealed by the high-resolution ω/2θ X-ray diffraction. A flat and sharp interface between LaMnO_3_ and SrTiO_3_ was verified by ADF−STEM imaging in combination with electronic energy loss spectroscopy. Anisotropic strain states were observed for the LaMnO_3_ film with no apparent in-plane strain and significant out-of-plane strain states. The B-site atomic displacement field displays a preferred position indicating the polarity of the LaMnO_3_ film. These findings open up a new way to fabricate complex oxide heterostructures with a well-defined interface using the one-stage hydrothermal route. Further work will include the investigation of the film formation process and morphology changes. Characterization of the magnetic properties of the epitaxial LaMnO_3_ film on the SrTiO_3_ substrate is in progress.

## Figures and Tables

**Figure 1 materials-15-05928-f001:**
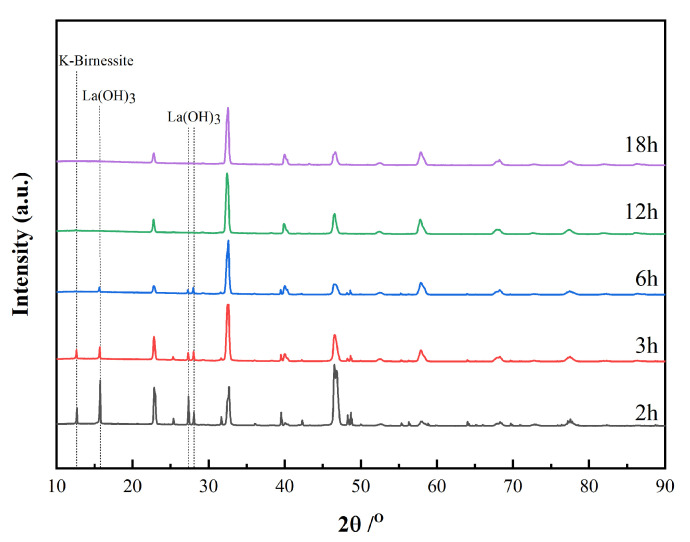
XRD patterns of LaMnO_3_ powders for periods of 2, 3, 6, 12 and 18 h. Reflections due to K-rich birnessite and La(OH)_3_ are indicated by dotted lines.

**Figure 2 materials-15-05928-f002:**
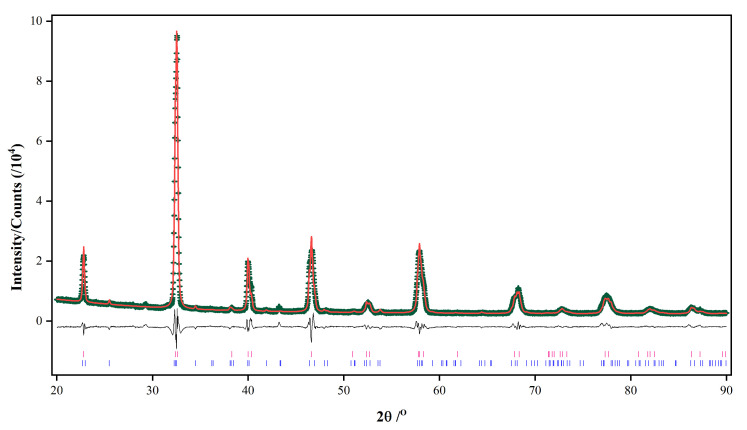
Rietveld phase quantitative analysis of the LaMnO_3_ powder at 18 h. The experimental powder X-ray diffraction data (black + marks), calculated powder X-ray diffraction data (black solid line) and the difference between experimental and calculated data (black lower line) are shown. Tick marks the reflection positions for trigonal (red, upper) and orthorhombic (blue, lower) LaMnO_3_.

**Figure 3 materials-15-05928-f003:**
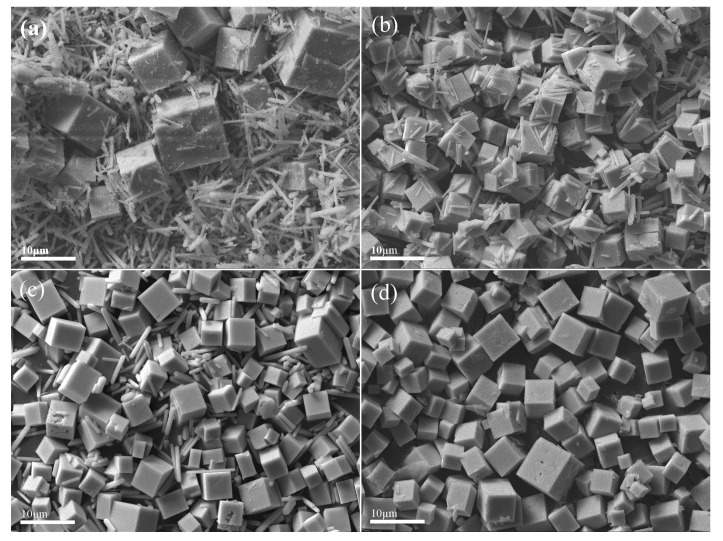
SEM images of products at different hydrothermal synthesis stages: (**a**) 2 h, (**b**) 3 h, (**c**) 6 h, (**d**) 12 h.

**Figure 4 materials-15-05928-f004:**
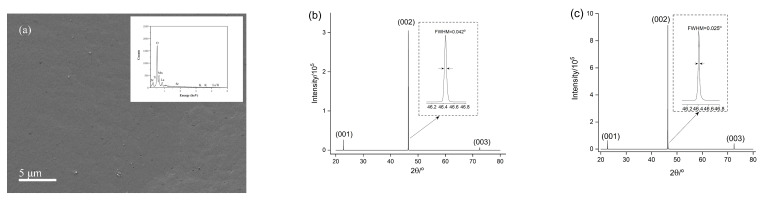
Characterization of the LaMnO_3_ film on the SrTiO_3_ substrate: (**a**) SEM image showing uniform and flat film surface. Inset: EDS spectrum; (**b**) symmetric ω/2θ scan of the LaMnO3 film. Inset: FWHM of the LaMnO_3_ film (002) peak; (**c**) symmetric ω/2θ scan of the SrTiO_3_ substrate. Inset: FWHM of the substrate (002) peak.

**Figure 5 materials-15-05928-f005:**
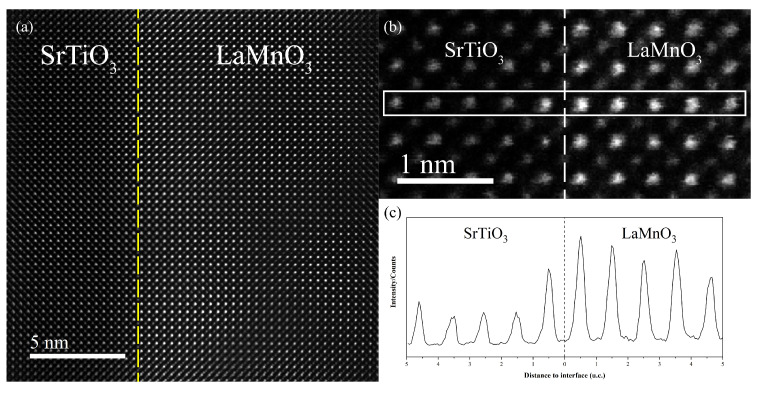
(**a**) ADF-STEM images of the interfaces between LaMnO_3_ films and the SrTiO_3_ substrate. Yellow dotted lines indicate the positions of the atomically sharp interface; (**b**) ADF−STEM images of the cross-section of LaMnO_3_ films with five unit cells of thickness in the [100] direction; (**c**) pixel in-tensity profile of an ADF line scan.

**Figure 6 materials-15-05928-f006:**
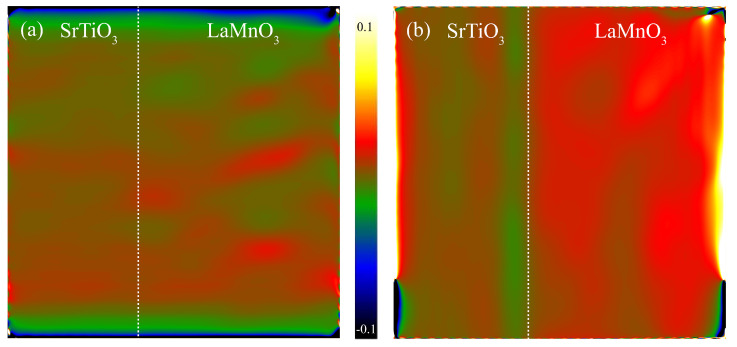
Geometrical phase analysis of LaMnO_3_ film showing in-plane (**a**) and out-of-plane (**b**) strain states.

**Figure 7 materials-15-05928-f007:**
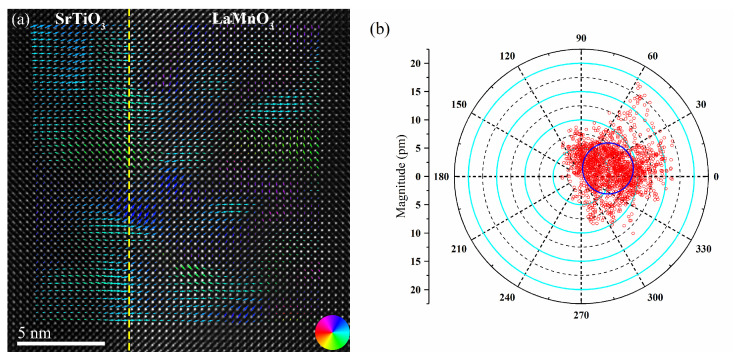
(**a**) ADF−STEM image with superimposed arrows representing the displacement of the Mn atoms with respect to the center of four neighboring A-site atoms. (**b**) The polar plot of the displacement of B-site atoms.

**Figure 8 materials-15-05928-f008:**
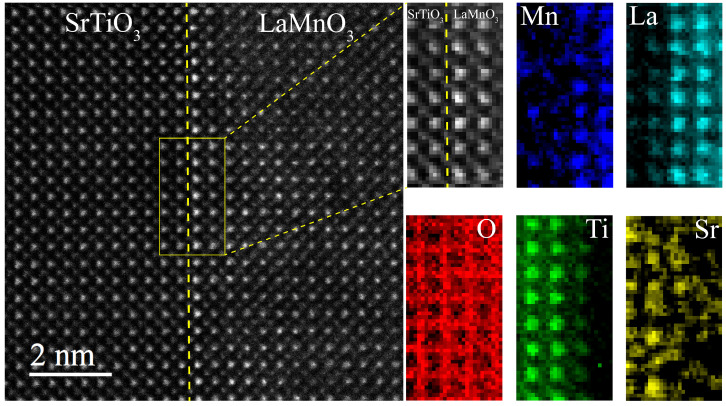
ADF−STEM images of the LaMnO_3_/SrTiO_3_ heterostructure with selected area for core-loss EELS mapping (outlined by a yellow rectangle) and the resulting map showing the spatial distribution of Mn (blue), La (cyan), O (red), Ti (green) and Sr (yellow) atoms.

**Figure 9 materials-15-05928-f009:**
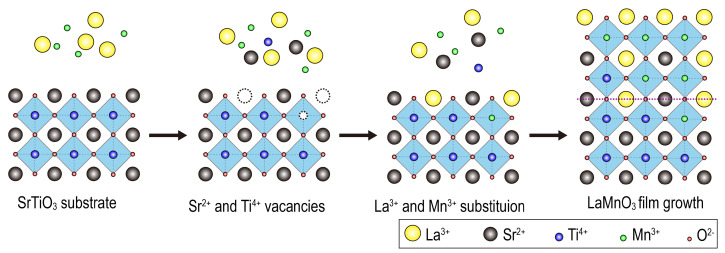
Schematic view of LaMnO_3_ epitaxial thin film growth mechanism.

**Table 1 materials-15-05928-t001:** Phase compositions of the hydrothermal products at different stages.

Time	Trigonal LaMnO_3_/%	Orthorhombic LaMnO_3_/%	K-Rich Birnessite/%	La(OH)_3_/%	Trigonal/ Orthorhombic
3 h	62.5	23.9	7.1	6.4	2.61
6 h	61.4	33.1	-	5.5	1.85
12 h	63.7	36.3	-	-	1.75
18 h	61.7	38.3	-	-	1.61

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
