# Peer review of "One-Stage Hydrothermal Growth and Characterization of Epitaxial LaMnO3 Films on SrTiO3 Substrate"

_materials, 2022, doi:10.3390/ma15175928_

Round 1

Reviewer 1 Report

Manuscript Title: One-step hydrothermal growth and characterization of epitax- 2 ial LaMnO3 films on SrTiO3 substrate

Manuscript ID: Materials-1876678

The paper is an interesting study of using a one-step hydrothermal route from La(NO3)3, MnCl2 and KMnO4 in an aqueous solution to make grown Epitaxial LaMnO3 thin films on SrTiO3 substrate.

The fabrication and characterization of the sample that would be useful for a better understanding of the performance of the system is satisfactorily presented.

Overall, the paper presents good findings and fits the journal scope, but with some minor revisions. The reviewer has the following specific comments that the authors might consider in the revision:

  1. In lines 22-23, please include the full names of VOC & SOFC for the readers. Also, there are more abbreviated words in the text that are required to be addressed as well like PLD, CVD and MBE in line 50.

  1. In line 136, there is no reference for the following sentence: “Previous research suggested that trigonal phase can be stabilized by improving the Mn4+ content in LaMnO3.”

  1. Figure 2 doesn’t have any legend.

  1. The quality of the images in Figure 4 a, b, and c are very low, especially their insets.

  1. In Figure 5a, SrTiO3 and LaMnO3 are not clearly visible. It’s better to use another color.

  1. Figure 5b doesn’t have any scale bar.

  1. The quality of Figure 5c is very poor.

Reviewer 2 Report

This manuscript reports the epitaxial LaMnO3 thin films growth on SrTiO3 substrate using a one-step hydrothermal route and the evolution of the growth mechanisms, nature of the interface between the LaMnO3 and the SrTiO3 were analyzed and presented. The manuscript is well written. This can be improved even further. I suggest to incorporate the following information in your manuscript.

Was the thickness of the films were measured or estimated? Please provide explanation.

Is this method of film deposition is scalable? I.e. can we able to get large area films?

Please provide the information on the thickness of the substrate used in the experiment.

Why specifically SrTiO3 was used as a substrate? Lattice matching condition for the film growth? Or is related to thermal expansion coefficient of films and the substrate? Please elaborate on these issues.

What was the volume of the autoclave?

Why the following step was employed during the hydrothermal growth?

“Powder samples were 86 removed after a certain time of period during the hydrothermal reaction at 340 °C.”

Please define all acronyms. ADF and VOC  missing. Please fix such problems throughout the article.

Reviewer 3 Report

The article is devoted to the study of the structural features of LaMnO3 thin films grown on a SrTiO3 substrate using a one-stage hydrothermal method. The proposed method for obtaining such structures is fairly well known, but the authors have made a number of improvements that make it possible to obtain sufficiently ordered structures. This work has a certain level of novelty and practical significance, and also corresponds to the subject of the declared journal. However, before accepting an article for publication, the authors should answer a number of questions that arose during its analysis.

1. The authors should explain the reason for choosing SrTiO3 as a substrate for the growth of the proposed thin films, why other more traditional materials for the synthesis of thin films were not used.

2. X-ray phase analysis data require significant improvements in the analysis of the obtained data and their interpretation. First, the authors should present the results of estimating the degree of crystallinity of the samples under study depending on the deposition time, as well as changes in the crystal lattice parameters and crystallite sizes. Secondly, a more detailed representation of the formation of diffraction reflections of the desired phases should be given, it is desirable to present selected areas on the diffraction patterns to reflect these changes.

3. When analyzing the morphology of thin films, the authors should explain what exactly such a rhomboid or cubic shape of grains from which films are formed is connected with, how these structures are interconnected and whether they are resistant to mechanical stress.

4. Have the authors carried out mechanical tests of the structures under study, if not, then the authors should, if possible, provide data on the strength and hardness of thin films.

5. In conclusion, the authors should give a more detailed explanation of further plans for this study and the prospects of the synthesized structures.

Round 2

Reviewer 3 Report

The authors answered all the questions, the article can be accepted for publication.